# Reproducibility of "Pixel-wise Anomaly Detection in Complex Driving Scenes " for ML Reproducibility Challenge 2021

## 1     Reproducibility Summary

2 The following paper is a reproducibility report for **Pixel-wise Anomaly Detection in Complex Driving Scenes**[4]
3 published in CVPR 2021 as part of the ML Reproducibility Challenge 2021. We reproduced the results quantitatively,
4 performed ablation studies, re-implemented the model in PyTorch Lightning and integrated WandB[1].
5 **Links:** PyTorch Repository, PyTorch Lightning Repository, & Our WandB Report.

### Scope of Reproducibility

7 Our efforts are focused on validating the authors' proposed anomaly segmentation framework, which employs the latest
8 re-synthesis approaches[8] and extends them to incorporate the advantages of uncertainty estimation methods[6] [7][11].
9 This proposed model outperforms existing re-synthesis techniques by a significant margin on the task of anomaly
10 segmentation on the Fishyscapes dataset.

### Methodology

12 We initially re-implemented the dissimilarity module in PyTorch Lightning using the authors' publicly available
13 source code. PyTorch Lightning increases the readability, reproducibility, and robustness of the code. It also provides
14 distributed training. We used pre-trained weights for image segmentation [14], image reconstruction[9] [13] and trained
15 the dissimilarity model on the Cityscapes dataset[3]. We trained all the models on a single P-100 GPU offered by
16 Kaggle for over 850 training hours.

### Results

18 Overall, our results back the original paper's claims. As shown in Table 3, our model outperforms the original study on
19 a few metrics but slightly falls behind on others on the benchmarked Fishyscapes dataset; discussed in section 5.1.

### What was easy

21 The paper was well-written and easy to understand. The provided open-source code is well-structured and modular.
22 Having pre-trained weights available for standard segmentation and reconstruction models reduced computational load.

### What was difficult

24 Even with modular code available, re-implementing the code in PyTorch Lightning proved more challenging than
25 expected. Our experiments were limited by the model's computational constraints, with an average training duration of
26 25 hours per model and kaggle only providing 9 hours of continuous training time. The data generation method is not
27 specified in the original repository; We have created a single script in our repository for the same.

### Communication with original authors

29 Authors were contacted via email to help clarify queries about the code, dataset generation, and discrepancies in the
30 results. Our final report received a positive review from the authors.

# 1  Introduction

Deep neural networks are becoming increasingly prominent in various applications, including computer vision, speech recognition, and language modeling. One such application is Semantic Segmentation, the per-pixel categorization of objects in an image. Recent advancements in Deep learning have brought significant improvement in this task, providing very accurate results, yet these networks fail to detect objects which are not part of the training dataset. Since anomalies are a component of many critical real-world applications, such as autonomous driving, implementing these networks for real-time situations necessitates overcoming issues like recognizing anomalous objects (any impediment on the road) and misclassification.

The existing methodologies either rely on predicted segmentation maps and their confidence scores or compare the reconstructed image to the predicted segmentation map to detect anomalies. However, these methods may fail as the segmentation network might make noisy anomaly predictions. Building over the existing resynthesis methods, the authors use uncertainty measures [6] [7] [11] to assist the dissimilarity network in differentiating the input and generated images that successfully generalizes to all anomalies.

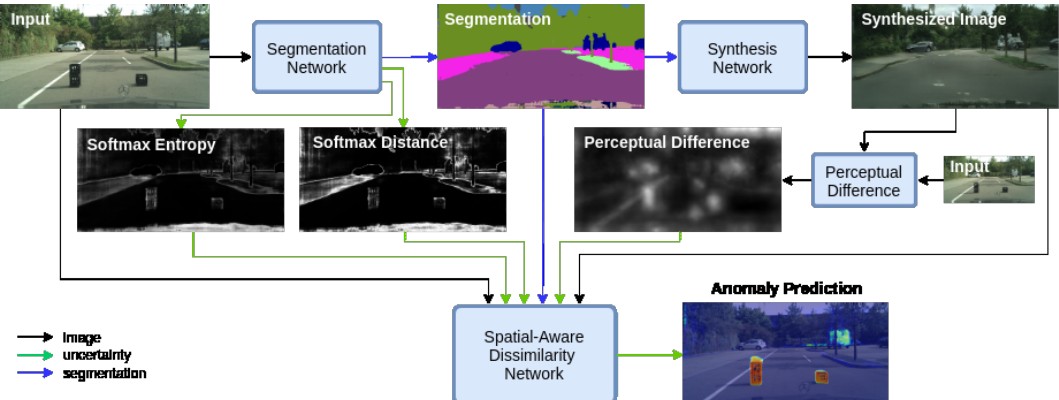

Figure 1: **Anomaly Segmentation Framework** described in **Section 3.2**

# 2  Scope of reproducibility

We sought to answer the following questions by reproducing the results of the paper, as well as conducting additional experiments to corroborate the paper's primary argument:

1. What is the importance of the uncertainty map's softmax entropy $H(1)$, softmax distance $D(2)$, and perceptual difference $V(3)$ (section 4.2.1)?
2. Can the network generalize to other segmentation and resynthesis networks (section 4.1.3)?
3. How does the performance of segmentation and resynthesis networks affect the overall performance of the model (section 4.2.2)?
4. Does the additional data generator added in this method improve the results (section 4.1.1)?
5. Is it better to choose weights for uncertainty maps for prediction during training or at the end through grid search (section 4.1.2)?
6. We further evaluate the sensitivity of our model to changes in the feature extractor module (encoder), activation function, and learning rate (section 4.2.3).

# 3  Methodology

## 3.1  Model descriptions

The model mainly has three modules namely segmentation, synthesis, dissimilarity and an ensemble:

**Segmentation Module :** We employ the pre-trained weights of the model as trained in [14] on Cityscapes dataset. In addition to generating a segmented image, we generate two dispersion maps, softmax entropy $H$ and softmax distance $D$, which prove beneficial in understanding anomalies within the generated segmentation map (p(c) is the softmax probability for class c). For each pixel x, $H$ and $D$ are calculated as follows:

$$H_x = - \sum_{c \in classes} p(c) \, log_2 \, p(c) \tag{1}$$

$$D_x = 1 - \max_{c \in classes} p(c) + \max_{c^1 \in classes \setminus (arg \, max_c \, p(c))} p(c^1) \tag{2}$$

**Synthesis Module :** To build a realistic image out of the segmentation map, we employ pre-trained weights from the model trained on Cityscapes dataset as a conditional generative adversarial network (c-GAN) [9] [13]. However, because the semantic map lacks information such as color appearance, per-pixel value comparison between the original input and the synthesized image is not possible. As a result, we use perceptual difference, which employs a pre-trained VGG16 model as a feature extractor to compare overall spatial structure rather than features such as color and texture, allowing us to better classify anomalies. For every pixel x of the input image and corresponding pixel r from the synthesized image $V$ is defined as follows :

$$V(x, r) = \sum_{1}^{N} \frac{1}{M_i} ||F^i(x) - F^i(r)||_1 \tag{3}$$

F(i) denotes the i-th layer with Mi elements of the VGG network and N layers, normalized between [0, 1].

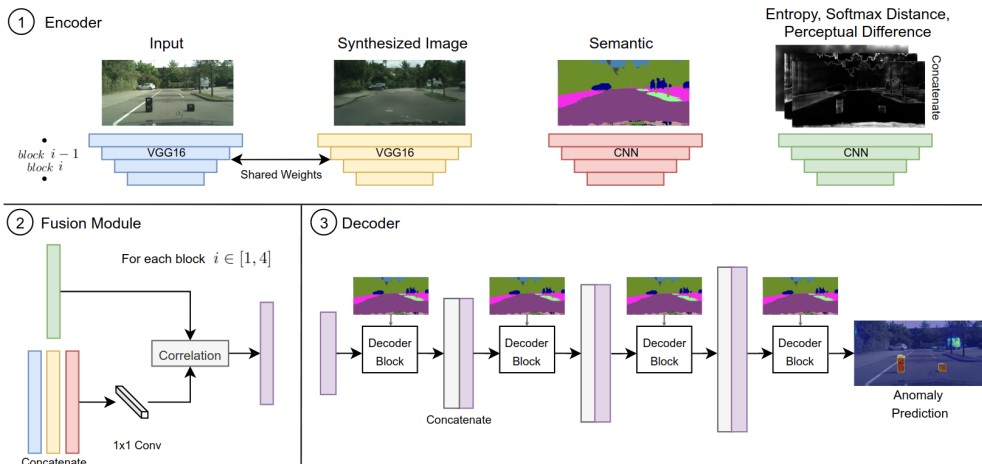

**Figure 2: Spatial-Aware Dissimilarity Module** described in **Section 3.1**

**Spatial-Aware Dissimilarity Module :** This module takes as input the original image, generated image, semantic map, and uncertainty maps (softmax entropy, softmax distance, perceptual difference) calculated in the previous steps to predict the anomaly segmentation map. It is mainly divided into three modules namely encoder, fusion and decoder:

1. **Encoder :** Encoder uses a pretrained VGG 16[12] to extract features of resynthesis and input image. Whereas a simple CNN is used to extract features from the uncertainty maps and semantic map.

2. **Fusion Module :** Concatenates and passes features extracted from resynthesis, input, segmented maps through a 1×1 convolution which is then passed into correlation block along with encoded uncertainty map where pointwise correlation is performed outputting four feature map resolutions corresponding to each of the four layers of the decoder.

3. **Decoder :** There are four decoder blocks used in the dissimilarity network. The first decoder block takes the lowest resolution feature map. The concatenation of the feature map from the fusion module and the output of the preceding decoder block is used as the input for all subsequent decoder blocks.

**Ensemble :** Dissimilarity module predictions are usually overconfident, which is reduced by combining final predictions with uncertainty maps (1),(2) ,(3) using appropriate weights obtained by grid search.

### 3.2 Training Procedure

We collect semantic maps and two uncertainty maps($H$ and $D$) by passing input images through the segmentation network. The synthesis network subsequently processes the predicted semantic map, which results in a photo-realistic

image. The perceptual difference is then calculated by comparing features between the input and generated images. The spatial-aware dissimilarity module receives uncertainty maps (1),(2) ,(3), input image, semantic map and resynthesized image to generate the anomaly prediction, which is then combined with uncertainty maps using optimal weights found via grid search.

### 3.3 Datasets

We employed the data generator technique provided in [8], which was expanded by assuming void labels in the ground truth semantic map as anomalies.

Evaluation is done on two sets of Fishyscapes Benchmark[2]:

1. **FS Lost & Found[10]:** A collection of 275 images with small objects (e.g. toys, boxes) on the roadway.

2. **FS Static[5]:** A collection of 1000 images from Cityscapes blended with anomalous Pascal VOC objects.

### 3.4 Hyperparameters

| Hyperparameter | Value |
|---|---|
| Number of epochs | 25 or 50 |
| Learning rate | 1E-4 |
| Learning rate Policy | ReduceLROnPlateau |
| Weight decay | 0.0000 |
| Power | 0.9 |
| Patience | 10 |
| Batch Size | 4 |
| $\beta 1$ | 0.5 |
| $\beta 2$ | 0.999 |

**Table 1:** We observed that the optimal model is obtained between 5 and 15 epochs. As a result, for additional experiments, we just train for 25 epochs.

### 3.5 Experimental setup

The training code was run on KAGGLE with Tesla P100-PCIE-16GB GPU (NVIDIA-SMI 450.119.04, Driver Version: 450.119.04, CUDA Version: 11.0).

### 3.6 Computational requirements

Using the pre-trained weights, the generation of semantic maps (softmax entropy, softmax distance) and synthesised images (along with perceptual difference) took 3 hours and 2 hours, respectively.

| | |
|---|---|
| Training time per epoch (Dissimilarity Module) | 30 minutes |
| GPU Requirement | 9 - 10 Gb |
| CPU memory Requirement | 2-3 Gb |
| Inference time (Dissimilarity Module) | 63 ms |

**Table 2:** Computational Requirement

## 4 Results

### 4.1 Results reproducing original paper

#### 4.1.1 Main Results

| Sample | FS L&F | | FS Static | |
|---|---|---|---|---|
| | ↑AP | ↓FPR95 | ↑AP | ↓FPR95 |
| Original Study | 43.22 | **15.79** | **72.59** | 18.75 |
| Reproduced Results | **44.47** | 18.7 | 71 | **17.17** |

**Table 3:** Benchmarked on Fishyscapes dataset[2] through model submission on official website

Table 3 shows that our model outperforms the original study in FPR 95 (False Positive Rate at 95% True Positive Rate) of FS STATIC and AP (Average Precision) of FS L&F on the benchmarked fishyscapes dataset, but slightly falls behind on AP of FS STATIC and FPR 95 of FS L&F; discussed in section 5.1

| Method | FS L&F (OURS) | | FS L&F | | FS Static (OURS) | | FS Static | |
|---|---|---|---|---|---|---|---|---|
| | ↑AP | ↓FPR95 | ↑AP | ↓FPR95 | ↑AP | ↓FPR95 | ↑AP | ↓FPR95 |
| Full Framework | 51 ± 7 | 41 ± 1 | 55 ± 5 | 40 ± 5 | 57 ± 4 | 28 ± 2 | 62 ± 5 | 26 ± 1 |
| w/o ensemble | 59 ± 5 | 61 ± 2 | 58 ± 9 | 66 ± 8 | 55 ± 6 | 32 ± 4 | 57 ± 6 | 41 ± 12 |
| w/o unc. maps | 28 ± 6 | 60 ± 8 | 39 ± 9 | 64 ± 10 | 28 ± 6 | 64 ± 4 | 38 ± 8 | 51 ± 4 |
| w/o data generator & w/o unc. maps | 14 ± 5 | 55 ± 9 | 15 ± 5 | 63 ± 12 | 11 ± 3 | 58 ± 9 | 14 ± 4 | 57 ± 11 |

**Table 4:** The following results are calculated on publicly available Fishyscapes validation datasets

The above results are reported as an average and standard deviation across five random weight initializations, as done in the original paper. Below are the training approaches whose results have been provided in Table 4:

1. **w/o unc maps :** Training performed without the use of uncertainty maps (1),(2) ,(3).
2. **w/o data generator & w/o unc. maps :** Training performed without using uncertainty maps (1), (2), (3) and without additional data generator as explained in section 3.3.
3. **Full Framework:** Full training as instructed in section 3.2 along with data generator.
4. **w/o ensemble:** Trained as instructed in section 3.2 except the ensemble.

Table 4 shows that most of the results are within the authors' given range. The minor differences can be explained since the model adjusts for larger variances shown in the results, and the five runs performed may not be adequate to generalize the model results. We can also see that uncertainty maps, data generator help to improve overall performance.

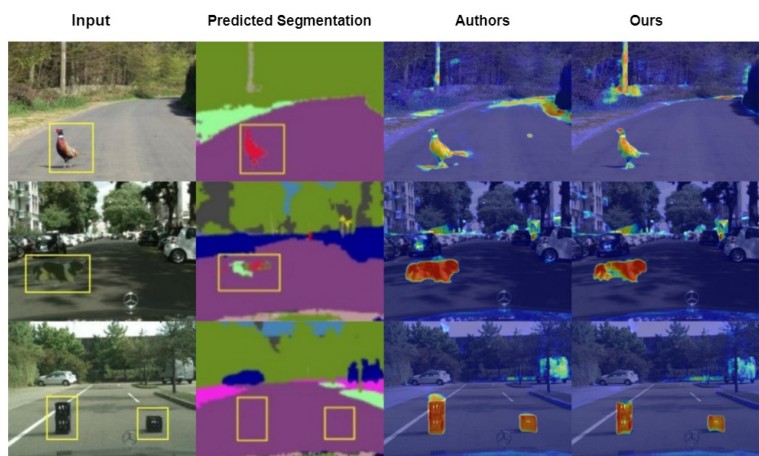

**Figure 3:** Our predictions are comparable with the authors' in detecting the main anomaly *

### 4.1.2 End to End ensemble Vs Ensemble Grid Search

| Method | FS L&F | | FS Static | |
|---|---|---|---|---|
| | ↑AP | ↓FPR95 | ↑AP | ↓FPR95 |
| ensemble grid search | 51.54 | **42.36** | 57.36 | **28.22** |
| end to end ensemble (**OURS**) | **61.01** | 63.38 | 47.02 | 46.31 |
| end to end ensemble | 59.6 | 58.6 | **61.1** | 37.3 |

**Table 5:** Comparison between end to end ensemble training and ensemble through grid search.

Below are the training approaches whose results have been provided in the above table:

1. **ensemble through grid search:** Training is carried out as stated in section 3.2, with weights for uncertainty determined at the end of training using grid search.
2. **end to end ensemble:** Here, weights for uncertainty maps are learnable parameters optimized during training.

---

*More images comparing author's outputs and ours are provided in the Supplementary material

We attribute higher deviation from results again to the fact that the experiments have a higher standard deviation, and even five runs may not be sufficient to generalize. However, in FS Static, our results show that an ensemble with empirical weights is more likely to identify intentionally blended objects, whereas end-to-end training is less likely since the network improves the weights before the final prediction. The increase in AP of FS L& F in end to end ensemble is expected as the network optimizes the weights efficiently. Our results supports the authors' contention that utilising an ensemble through grid search yields lower FPR 95 values. In safety-critical contexts, this makes the ensemble with empirical weights for final prediction more practical (e.g., autonomous driving).

### 4.1.3 Generalizability of the model

| Method | FS L&F | | FS Static | |
|---|---|---|---|---|
| | ↑AP | ↓FPR95 | ↑AP | ↓FPR95 |
| Original | **51.54** | **42.36** | **57.36** | **28.22** |
| Light (**OURS**) | 32.88 | 50.96 | 28.68 | 31.84 |
| Light | 36 | 46.4 | 33.4 | 36.1 |
| Im. Resyn.++ | 5.7 | 47.7 | 8 | 62.7 |

**Table 6:** Testing dissimilarity module on lighter segmentation and resynthesis frameworks.

Below are the combination of segmentation and resynthesis networks used, whose results have been shown in Table 6:

1. **Original** : SDCNet Segmentation + C-GAN Resynthesis Networks.
2. **LIGHT** : ICNet Segmentation + Spade Resynthesis Networks.
3. **Im. Resyn.++** : Image Resynthesis++ is used for comparison as our method builds upon it.

According to Table 6, the results for the light framework are significantly lower than the original model but significantly better than the original image resynthesis method(Im. Resyn.++), proving the authors' claim that this method generalises well enough to be used as a wrapper to already existing segmentation and resynthesis networks.

## 4.2 Results beyond original paper

### 4.2.1 Importance of Uncertainty Maps

| Method | FS L&F | | FS Static | |
|---|---|---|---|---|
| | ↑AP | ↓FPR95 | ↑AP | ↓FPR95 |
| Full Framework | 51.54 | 42.36 | **57.36** | 28.22 |
| W/O Softmax Entropy | 37.03 | 42.8 | 40.17 | 35.9 |
| W/O Softmax Distance | 45.93 | 41.47 | 53.47 | **27.47** |
| W/O Perceptual Difference | **51.7** | **39.57** | 51.8 | 27.83 |

**Table 7:** Assessing the significance of each uncertainty map

From Table 7, we can observe that omitting Softmax Entropy lowers all outcomes. Even Softmax Distance has a noticeable impact on the model's performance. However, removing Perceptual Difference has almost no effect (except on AP of FS STATIC), which we believe is because the resynthesized image is already passed to the dissimilarity module, which does the task of differentiating input and resynthesized image.

### 4.2.2 Importance of performance of Segmentation and Resynthesis networks

| Method | FS L&F | | FS Static | |
|---|---|---|---|---|
| | ↑AP | ↓FPR95 | ↑AP | ↓FPR95 |
| Moderate1 | 44.767 | **38.88** | 31.6 | 39.167 |
| Moderate2 | 49.87 | 43.83 | 50.93 | 29.2 |

**Table 8:** Effect of segmentation and resynthesis networks.

Below are the combination of segmentation and resynthesis networks used, whose results have been shown in Table 8:

1. **Moderate1** : SDCNet Segmentation + Spade Resynthesis Networks.
2. **Moderate2** : ICNet Segmentation + C-GAN Resynthesis Networks.

From Table 8, the quality of the segmentation and resynthesis networks has an impact on the outcomes, and it is clear that the original performs better than light because it employs superior segmentation and synthesis methods. Results of Light version and original can be referenced from Table 6. Original findings outperform Moderate1 (lighter segmentation, same resynthesis network) and Moderate2 (same segmentation, lighter resynthesis network), demonstrating that both synthesis and segmentation are critical components of the model and are directly connected to its overall performance.

### 4.2.3 Network Tuning

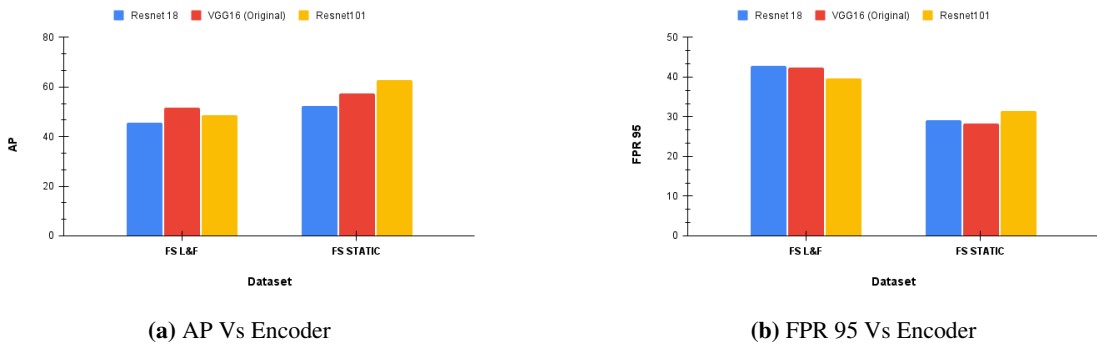

(a) AP Vs Encoder       (b) FPR 95 Vs Encoder

**Figure 4:** VGG16 dominates ResNet18 & ResNet101 dominates VGG16 in FPR 95 of FS L&F and AP of FS STATIC

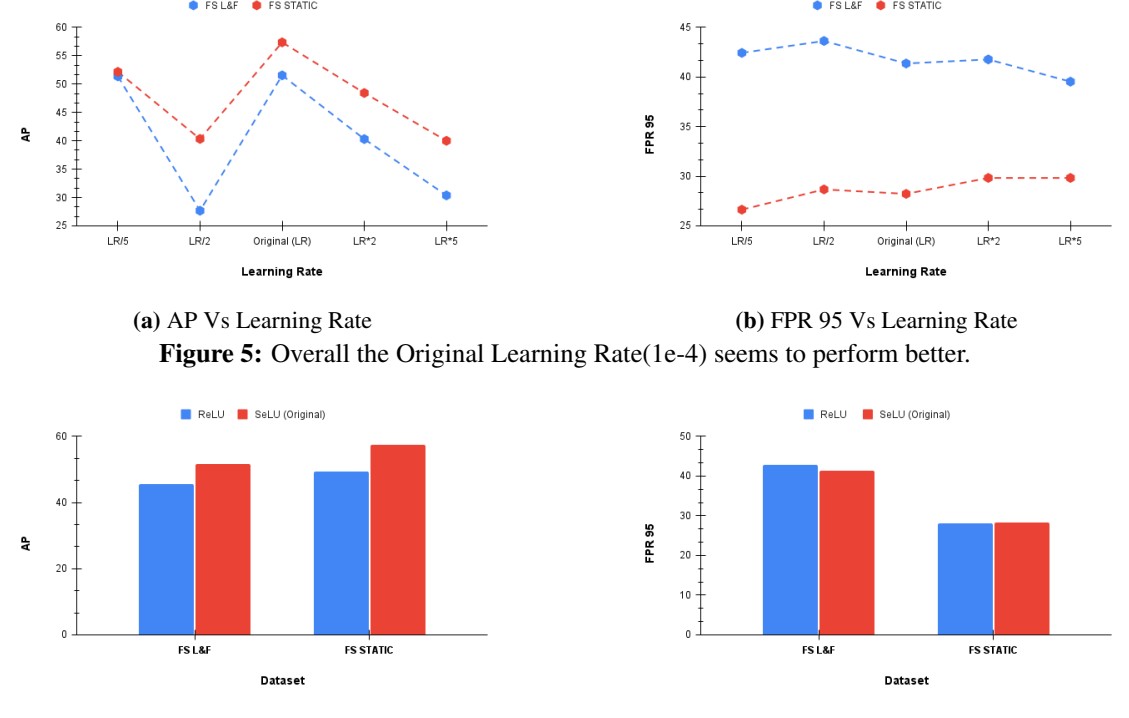

(a) AP Vs Learning Rate       (b) FPR 95 Vs Learning Rate

**Figure 5:** Overall the Original Learning Rate(1e-4) seems to perform better.

(a) AP Vs Activation Function       (b) FPR 95 Vs Activation Function

**Figure 6:** We can observe the supremacy of SeLU over ReLU when used in dissimilarity module in all the outcomes.

## 5  Discussion

Our results are on par with those of the authors' on the private test data but do not exactly match the results on the validation datasets despite utilizing the exact parameters described in the paper. Nonetheless, they are practically on par with them, proving the authors' claim. Section 4.1.3 demonstrates that the model can generalize to even lower-level segmentation and synthesis networks, allowing it to be used as a wrapper to any pre-trained segmentation and resynthesis models. Section 4.2.2 demonstrates the importance of segmentation and resynthesis network performance. In Section 4.1.2, we can see that ensemble using grid search outperforms end-to-end ensemble.

We provide two arguments to explain why our results differ from the authors. Firstly with five random initializations, we can see very high standard deviation; so, other standard deviation remains to be accounted for, and these runs are insufficient to generalize conclusions. Secondly, the weights for FS L&F and FS STATIC for uncertainty maps after ensemble are different, as mentioned in section 5.1.

We quantitatively confirmed the relevance of uncertainty maps in section 4.2.1, where we observed that removing the softmax entropy map or the softmax distance map has a considerable effect on outcomes, although removing the perceptual difference had little to no effect. To improve anomaly segmentation, all uncertainty maps are heavily employed in conjunction with their complimentary information.

We also attempted to enhance the findings in section 4.2.3; (4a, 4b) we replaced the VGG16 encoder with ResNet18 and ResNet101; (6a, 6b) We tried commonly used ReLU instead of SeLU activation function in the dissimilarity module; (5a, 5b) we modified learning rates. Despite improvements in a few outcomes, the parameters used by authors proved to be the best, with a strong balance among all metrics.

## 5.1 Further discussion on discrepancies of the results

Table 4 shows that ensemble weights are an important aspect of the model, as they reduce the model's overconfidence. The challenge, however, is determining the optimal ensemble weights for the model. The sensitivity to these ensemble weights is one of the key causes for our results' divergence.

For most of our runs, the ensemble weights for FS L&F differed from those for FS STATIC. This makes keeping the final ensemble weights a little more challenging. In most circumstances, [0.75,0.25,0,0] (the four weights being for original output, entropy, perceptual difference, and softmax distance, respectively) yields a better outcome. However, employing these weights degrades the results of FS L&F; which can be seen from its FPR 95 in Table 3.

Table 4 shows that the model accounts for more significant variations, and the five runs done may be insufficient to generalise the results. This large deviation in model results might be the reason for lower AP of FS Static in Table 3.

Another thing to note is that all of the tuning and validation is done on the Fishyscapes validation datasets, even though the Fishyscapes paper expressly warns that this may result in overfitting to the validation set and will most likely not result in better performance on the unexpected test data. The fact that our model doesn't perform on par with the original model on the validation datasets, but performs well on private test data proves this claim. But owing to a shortage of suitable anomaly detection datasets we are forced to use these validation datasets for tuning. Furthermore, choosing tuning/validation datasets for anomaly segmentation is always difficult since we are constantly in danger of overfitting the parameters to specific abnormalities.

**What was easy**

The paper was well-written and easy to understand. The provided open-source code is well-structured and modular. Having pre-trained weights available for standard segmentation and reconstruction models reduced computational load.

**What was difficult**

Even with modular code available, re-implementing the code in PyTorch Lightning proved more challenging than expected. Our experiments were limited by the model's computational constraints, with an average training duration of 25 hours per model and kaggle only providing 9 hours of continuous training time. The data generation method is not specified in the original repository; We have created a single script in our repository for the same.

**Communication with original authors**

Authors were contacted via email to help clarify queries about the code, dataset generation, and discrepancies in the results. Our final report received a positive review from the authors.

**Acknowledgement**

We thank the original authors for their quick and thorough response, facilitating reproducibility. We also thank Boyang SUN for assisting us with benchmarking results on the Fishyscapes website, making reproducibility complete.

**Future Scope**

From Table 4, we see that the dissimilarity module alone takes 63ms. Further, considering inference time of segmentation and resynthesis networks limits the model from applying it for real-time tasks. Hence improving the model for real-time performance would be a considerable task for future. One approach to get around this could be to use distributed training. Because PyTorch Lightning is hardware agnostic and easy to scale, it streamlines this task.

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
