# OpenReview forum: "Reproducibility of "Pixel-wise Anomaly Detection in Complex Driving Scenes" for ML Reproducibility Challenge 2021"
_ML_Reproducibility_Challenge/2021/Fall — Reject_

### Official Review · Reviewer_Rjnf · 2022-03-04
**Solid work but not surprising and there are issues with presentation**

**Rating:** 4
**Confidence:** 4

**Review:**

This is a well done reproducibility study in the literal sense of the word. Most of the experiments from the original paper are reproduced, some extra analysis is done. All this is a solid piece of work. It is obvious that this has been done quite carefully. However, after reading this report there is no feeling of anything extra done that adds something interesting and valuable to the community apart from the confirmation that the claims from the original paper have been confirmed.

That could have been a minor complain although there is a bigger issue with the report that prevents me from giving it a higher score - this is its presentation. Unfortunately, the presentation of the report can be largely improved, please see details below.

Specific comments/suggestions (in the order appearing in the text mostly, not in the order of importance):
1. Line 16. Missing reference to Fishyscapes dataset.
2. Line 13. "PyTorch Lightning increases the readability..." - rather unjustified claim
3. Line 27. "We have created a single script ..." - unclear. Is it important that this is a single script and not multiple? Also "for the same" is not the best wording
4. Lines 32-33. "Deep neural networks...", line 35 "yet these networks", lines 39-40 "The existing methodologies..." - it is better to add references to confirm these claims.
5. Line 41. "resynthesis methods" - since semantic segmentation has been defined before, it would be appropriate to define this as well.
6. Line 45. "the results of the paper" - missing reference
7. Lines 47-48. "softmax entropy", "softmax distance", and "perceptual difference" have not been defined yet
8. Line 49. Other than what?
9. Line 59. "has three modules" and 4 is listed later
10. Line 60. "on Cityscapes dataset" - missing reference plus it is better to specify explicitly whether it is the same as in the original paper or not.
11. Line 62. "which prove beneficial" -> "which are proved to be beneficial", proved by whom? missing reference
12. Lines 62-63. "p(c) is the softmax probability for class c" is better to move after "as follows"
13. Lines 67-69. How this is ensured to capture specifically spatial structure and not color or texture?
14. Figures 1, 2 and 3 are not referred to in the text.
15. Line 76. "a simple CNN" - what is this?
16. Line 94-95. "We employed..." - how does this compare to what have been dine in the original paper? What are "void labels"?
17. Section 3.4. Missing text. It is not enough to have a table (Table 1) that is not referred to in the text.
18. Table 1. What are Power, \beta_1, \beta_2. Why weight decay is 0.0000 - if it's 0, it may have less decimal digits, if it is not zero, the non-zero value should be stated. From caption - what is the optimal model?
19. Table 3. Caption should be a more full sentence.
20. Table 3 and later. FS L&F acronym is not defined.
21. What is the difference between "Full Framework" in Table 4 and results in Table 3?
22. Line 112. "above results" and line 124, "above table" - it is better to refer by table number.
23. Section 4.2.1. Some introduction would be appreciated.
24. Line 145. "Even Softmax Distance" - why even?
25. Section 4.2.3 - missing text.
26. Figure 4. Caption should be more full. Font of axis labels is too small
27. Line 167. "other standard deviation" - "other" is unclear
28. Lines 166-177. It is better to move these discussions to the corresponding sections with missing text.
29. Line 183. "[0.75, 0.25, 0, 0]" - where do these numbers come from?
30. Line 189. "expressly" -> "explicitly"?
31. Lines 195-205. Exact copy from the first page.



Minor:
1. Throughout the text. There are at least 3 different version how to write "Kaggle" in the text. It should be consistent.
2. Line 27. Either replace ";" with ".", or "We" should be lower case
3. Line 37. "real time" -> "real life"?
4. Line 42. "authors" - add after that "of the original study [4]"
6. Line 71. "Mi" -> "$M_i$"
7. Line 87. missing blank after "maps"
8. Line 90 and line 114 missing blank after "(1),"
9. Table 2 caption. RequirementS
10. Section 4. "AP of [dataset]" -> "AP on [dataset]"

---

### Meta-Review · Program_Chairs · 2022-04-07

**Recommendation:** Reject
**Confidence:** 5

**Metareview:**

While the current report does make an effort in reproducing the original paper, it does not bring any additional insights or contributions -- no analyses or ablation studies, no applications on other datasets. There are also significant presentation issues (as highlighted by reviewer Rjnf), which indicates that the report needs more polishing before acceptance.

---

### Decision · Program_Chairs · 2022-04-09

Reject